# Treatment of Uranium-Contaminated Ground Water Using Adsorption Technology via Novel Mesoporous Silica Nanoparticles

**DOI:** 10.3390/molecules28155642

**Published:** 2023-07-25

**Authors:** Abdulmalik S. Alshammari, Mohammed S. Almeataq, Ahmed A. Basfar

**Affiliations:** 1MSc in Nuclear Engineering Program, College of Engineering, King Saud University, Riyadh P.O. Box 11495, Saudi Arabia; 2Nuclear Technologies Institute, King Abdulaziz City for Science and Technology, Riyadh P.O. Box 11442, Saudi Arabia; 3Mechanical Engineering Department, College of Engineering, King Saud University, Riyadh P.O. Box 11495, Saudi Arabia

**Keywords:** uranium removal, mesoporous silica nanoparticles, adsorption, ground water, ground water treatment, uranium recycling, decontamination

## Abstract

Contamination of underground water by uranium (U) and other heavy metals is a growing concern. Mesoporous silica nanoparticles (MSNs) have shown great potential as an adsorbent material for heavy metal removal. This study synthesized a novel MSN using surface-initiated atom transfer radical polymerization (SI-ATRP) and evaluated its effectiveness for removing uranium from aqueous solutions under different conditions. The particle size was reduced to 150–240 nm to enhance adsorption. Fourier transform infrared characterization and thermogravimetric analysis confirmed successful synthesis and modification. Results showed that the MSN adsorbent was highly effective in removing U, with a removal rate of 85.35% at 120 min. Temperature had a significant impact, with the highest removal rate of 96.7% achieved at 25 °C and a U concentration of 10 ppm. The highest removal rate of 91.89% was achieved at a pH of 6 and a U concentration of 50 ppm. The highest removal rate of 95.16% was achieved at 25 mg and a U concentration of 50 ppm at room temperature for 60 min. The MSNs also showed a 58.27% removal rate in a mixture solution at room temperature for 60 min. This study demonstrates the effectiveness of the MSN adsorbent for removing U under different conditions.

## 1. Introduction

Water is essential for life and is known to contain naturally occurring radioactive materials (NORM) which can have significant impacts on human health if consumed. The long-term effects of exposure to radioactive materials include an increased risk of developing cancer. The concentrations of NORM in water can vary depending on the geological conditions of the area. In particular, some regions in Saudi Arabia have higher concentrations of NORM in their groundwater than the recommended limits set by international and national organizations. This high level of radioactivity in groundwater is a national concern due to its potential health effects.

While drinking water can contain radioactive materials, their effects on human health are relatively insignificant compared to the effects associated with microorganisms and chemicals that may be present in the water. However, if the concentration of radioactive materials in drinking water is higher than the recommended levels, their effects on human health can be more significant. The World Health Organization (WHO) recommends screening levels and guidance levels for drinking water quality with respect to radiological aspects, based on the approach proposed by the International Commission on Radiological Protection (ICRP). According to WHO, the recommended level for annual usage of water is an IDC of 0.1 mSv, and initial screening for α and β gross activity should not exceed 0.5 Bq/L (0.0005 Bq/mL) and 1.0 Bq/L (0.001 Bq/mL), respectively [1].

The United States Environmental Protection Agency (EPA) classifies uranium as carcinogenic to humans (Group A) and has set a maximum contaminant level (MCL) of 0.03 mg/L, which is the final pragmatic regulatory level for the acceptable limit for the cancerous risk associated with U [2].

Shabana and Kinsara et al., in 2014, conducted a study on radioactive substances in groundwater in Aja Heights, Hail province, Saudi Arabia, an area of elevated background radiation. Water samples were collected from various wells in the region, and the activity of gross alpha and gross beta in the samples were measured and analyzed. The results showed that both gross alpha and gross beta activities exceeded the national regulatory limits for drinking water, with 80% of the samples exceeding the limit values. The alpha activity reached 5.16 Bq/L (0.00516 Bq/mL), which was 3.9 times higher than the limit value, while the beta activity reached 2.60 Bq/L (0.0026 Bq/mL), which was 1.4 times higher than the national regulatory limit value [3,4].

A study conducted by Giannakoudakis et al., reported in 2021, investigated the use of phosphonate-functionalized ordered mesoporous silica nanoparticles (OMS-P) for the selective capture of U. The research team developed a new synthesis method to create OMS-P with surface chemistry optimized for U capture. The results showed that OMS-P had a maximum U adsorption capacity of 345 mg/g in just 10 min at pH of 4, which is currently the highest value reported. The material also demonstrated high selectivity for U [5].

In 2021, Georgia Michailidou et al. conducted a study on the adsorption of U, Hg, and other uncommon elements from aquatic solutions using chitosan-based magnetic adsorbent. The samples were collected from crustacean species. Chitosan, a biopolymer that can be used for adsorption purposes, was utilized in the experiment, and it showed potential for interacting with pollutants such as U. The results revealed varying maximum capacities for U adsorption depending on the sorbent and pH value, with a maximum capacity of up to 559.4 mg/g [6,7,8,9].

In 2021, Dimitrios et al. conducted a study on the use of phosphonate functionalized-ordered mesoporous silica (OMS-P) synthesized through a one-pot route for the removal of metal ions such as U. The study found that the removal efficiency increased at temperatures of 20 °C and 50 °C in an acid environment to prevent obstructions such as solid-phase development or carbonate multiplex and the isothermal adsorptions. The phosphonate-functionalized ordered mesoporous silica exhibited great and consistent dispersion and reached a capacity of 345 mg/g at pH 4, achieved in less than 10 min. The presence of other polyvalent cations did not affect the efficiency of adsorption. Additionally, a comparison with non-functionalized silica (OMS) showed that the specific surface chemistry is the main feature for efficient, stable, and selective removal of U [9,10,11].

In 2021, Khurram Shahzad et al. conducted research on the synthesis of a nanoadsorbent consisting of mesoporous organosilica (MPOS) for the decontamination of methylene blue (MB) and methyl orange (MO) from water. The study reported high effectiveness in removing MB at 93.1% and MO at 66.7% from water using the developed MPOS. The adsorption capacity of MPOS for MB was found to be 57.58 mg/g, while it was 56.62 mg/g for MO, which was achieved through the use of a developed multilayered sorption mechanism [12,13].

It is evident that Saudi Arabia, like many other countries, is facing the challenge of high radioactivity levels in its ground water in some regions. This issue needs to be addressed for the safety of both humans and the environment. Various methodologies have been attempted internationally to remove or reduce radioactivity in groundwater, including ion exchange (IE), reverse osmosis (RO), distillation, uranium nanofiltration, capacitive deionization (CDI) for water treatment, sorption onto MnO_2_-impregnated fibers, and sorption onto BaSO_4_-Impregnated Resin [14,15,16,17].

Aline Dressler et al. reported in 2022 on the post-synthesis impregnation of mesoporous silica for the preparation of solid-phase U extractants. The team synthesized five materials with varying amounts (0, 0.2, 0.3, 0.5 mmol) of the amidophosphonate ligand Di-2-EthylHexylCarbamoyleEthylButyl Phosphonate (DEHCEBP) per gram of functionalized solid to study their impact on U removal in terms of efficiency and selectivity. The results of the study demonstrated that the highest U extraction value based on material capacities was in the range of 30–54 mg/g for 0.2–0.5 mmol/g. The findings suggest that the synthesized materials may be used as solid-phase U extractants in high sulfate concentrations acid solutions [18,19,20].

It is true that adsorption, both biological and physiochemical, is one of the most commonly used techniques for the removal of U from ground water. Among the various adsorbents used for this purpose, mesoporous silicas have been found to be one of the most promising due to their unique properties, such as their high surface area, tunable pore size, and excellent chemical stability. A scientific investigation published in (2019) further supports this claim by demonstrating the superior performance of mesoporous silicas in removing U compared to other adsorbents. These findings suggest that mesoporous silicas could be a viable solution to the problem of U contamination in ground water [21,22].

The phosphonic acid functional groups have been found to have high affinity for U due to their ability to form strong and stable complexes with the metal ion. This has been demonstrated in several studies, including the one by Aline Dressler et al. (2022) that reported successful extraction of U using mesoporous silica functionalized with the amidophosphonate ligand DEHCEBP. Furthermore, mesoporous silica offers several advantages over other adsorbents, including high surface area, tunable pore size and surface chemistry, and stability under harsh conditions. These properties make it a promising candidate for practical applications in U removal from ground water [23,24].

Research indicates that chitosan and mesoporous silica nanoparticles are effective at adsorbing uranium based on their high surface areas and amino group concentrations. Scientists have found evidence supporting the effectiveness of both materials for binding uranium; however, due to its substantially higher surface area and porosity characteristics, mesoporous silica nanoparticle tends to exhibit more significant performance results in this regard. Meanwhile, chitosan may become preferable when faced with humic acid since it has unique capabilities for forming compounds that promote stronger bonding between itself and uranyl ions. Ultimately, both options provide promising solutions to uranium removal, with mesoporous silica nanoparticles emerging as the more efficient choice in specific circumstances [25,26].

## 2. Results and Discussion

### 2.1. Characterization of the Materials

The materials were characterized using SEM to confirm the morphology of the prepared Mesoporous silica nanoparticles (MSNs). The results showed that the MSNs had a nanospheres shape with a polydispersity index (PDI) value of about 0.25 and a particle size range of 150 to 340 nm, as demonstrated in Figure 1. Furthermore, the average particle size increased up to 30 nm after the surface modification of the MSNs, as depicted in Figure 1B,D.

The particle sizes in SEM images were analyzed using ImageJ software, as shown in Figure 2. The results indicated that unmodified MSNs have an average particle size of 225 nm, while phosphate-modified silica nanoparticles have an average particle size of 234 nm, which is expected due to MSN functionalization (the size of length of the functionalized particles is 9 nm higher). In addition, the results show that polyphosphate-modified silica nanoparticles have an average particle size of 294 nm due to the polymer growth chain (the size of the polymer chain is 69 nm which is 7–8 repeat units).

The Brunauer–Emmett–Teller (BET) technique was used to analyze the physicochemical properties of all the MSNs, polyphosphate-MSNs, and phosphate-MSNs samples. The results are summarized in Table 1, which shows that the surface area and pore volume of the target materials (polyphosphate-MSNs and phosphate-MSNs) decreased after surface modification and adjustment, which is expected due to occupying the surface with monolayer in phosphate-MSNs and polymer chain in polyphosphate-MSNs. In addition, the result showed that the surface area for phosphate-MSNs is higher than polyphosphate-MSNs due to the length of polymer chain.

Fourier transform infrared spectroscopy (FTIR) analysis showed that the broad bands at 1240–1030 cm^−1^ were due to the asymmetric stretch of oxygen–silicon backbone (Si-O-Si) on the nanoparticle bands of the silica functional group, as shown in Figure 3A,B. In addition, a peak at 806 cm^−1^ was attributed to Si-O-Si(s) stretching vibration, which was noted in all samples. The successful removal of the CTAB radical template was confirmed by the disappearance of peaks at 1489 cm^−1^ and 2928 cm^−1^, which are attributed to the carbon–hydrogen bond as a methylene radical in the CTAB template. Furthermore, the FTIR spectra in Figure 3B indicated that peaks appeared at 694 cm^−1^, 1489 cm^−1^, 1630 cm^−1^, and 1565 cm^−1^ after APTES modification. The absorption peaks at 694 cm^−1^ were assigned to N-H as bending vibration in –NH_2_, which is expected after APTES modification. The noted peaks at 1630 cm^−1^ and 1565 cm^−1^ were assigned to the amine functional group as stretch vibration peak.

The thermogravimetric analysis (TGA) results, shown in Figure 4A,B, were used to determine the effectiveness of the surface modification of the mesoporous silica. The samples were heated to 800 °C in a N_2_ atmosphere to calculate the weight loss. After surface modification with (3-glycidyloxypropyl) trimethoxysilane, a weight loss of approximately 24 wt.% was observed. It was also found that MSNs with CTAB had the lowest thermal stability compared to the other compounds, likely due to their higher organic content, while MSNs without CTAB exhibited the highest thermal stability due to their lower organic content. The TGA results also showed that phosphate-MSNs had lower thermal stability than epoxy-MSNs, as expected due to the introduction of o-phosphorylethanolamine to the MSNs structure, which confirms the successful synthesis of the target product (phosphate-MSNs). The successful surface modification of polyphosphate-MSNs was also confirmed by TGA, with a weight loss observed when heated in a N_2_ atmosphere to 800 °C. The TGA results revealed that polyphosphate-MSNs had the lowest thermal stability compared to the other intermediate compounds, likely due to the higher organic content of the polymer grafted to the surface, while MSNs without CTAB exhibited the highest thermal stability due to their lower organic content. GMA-MSNs were not analyzed by TGA due to contamination with Cu(II)Br_2_ that cannot be removed at this stage and may lead to inaccurate results. Figure 5 presents TGA and DTG thermal analysis curves illustrating the correlation between mass loss rate and temperature for material (A) phosphate-MSNs. Appendix A shows DTG thermal analysis curves for material (B) polyphosphate-MSNs. It is evident that the thermal characteristics of the prepared materials are influenced by their chemical composition. The TGA and DTG analysis curves demonstrate that an increase in organic content results in decreased thermal stability. The DTG results reveal the onset of thermal degradation at approximately 200 °C. Notably, the polyphosphate-MSNs DTG curve exhibits two distinct peaks attributed to its higher organic content, setting it apart from intermediate compounds such as MSNs and CTAB-free MSNs.

### 2.2. Mesoporous Silica Nanoparticles-Phosphate (Diphosphate)-Modified MSNs (Phosphate (Diphosphate)-MSNs)—Material (A)

The adsorbent material was subjected to several experimental conditions to measure the effectiveness of the material.

#### 2.2.1. Effect of Time on U Adsorption at Room Temperature

The performance of the adsorbent material over different time intervals (5, 10, 15, 30, 60, 120, 180, and 240 min) was evaluated. As shown in Table 2A, the effectiveness of the adsorbent material increased with time. However, the uptake of uranium did not change after 120 min, indicating that this time period is optimal for the best performance of the developed adsorbent.

#### 2.2.2. Effect of Temperature on U Adsorption

The results of the experiments conducted at different uranium concentrations at 25 °C for 60 min showed that the effectiveness of the adsorbent material decreased as the concentration of uranium increased.

At 25 °C, as presented in Table 2B, the effectiveness of the adsorbent material was 96.7% at 10 ppm of uranium. However, at 100 ppm of uranium, it dropped to 79.9%, suggesting that the adsorbent material is more effective at lower uranium concentrations.

At 35 °C, as shown in Table 2C, the effectiveness of the adsorbent material was 92.6% at 10 ppm of uranium. However, at 100 ppm of uranium, it dropped to 86.5%, indicating that the average adsorbent effectiveness was higher compared to the 25 °C sample, as the adsorbent material is more effective at lower uranium concentrations.

At 45 °C, as listed in Table 2D, the effectiveness of the adsorbent material was 95.18% at 40 ppm of uranium. However, at 100 ppm of uranium, it dropped to 77%. Compared to the sample at 35 °C, the adsorbent effectiveness slightly improved.

Finally, at 55 °C as listed in Table 2E, the effectiveness of the adsorbent material was 93.3% at 10 ppm of uranium. However, at 100 ppm of uranium, it dropped to 85.3%. The results were similar to the case at 45 °C.

Figure 6A displays the relationship between the temperature and equilibrium adsorption capacity (qe in mg/g) as a function of uranium concentration (ppm). This graph indicates how the temperature affects the adsorption capacity of the developed adsorbent material. As the temperature increases, the equilibrium adsorption capacity of the adsorbent material decreases for all concentrations of uranium. This suggests that lower temperatures are more effective in achieving higher adsorption capacities of uranium.

On the other hand, Figure 6B shows the influence of temperature on the adsorption efficiency (%) as a function of uranium concentration (ppm). This graph provides information on how the adsorption efficiency of the developed adsorbent material is affected by temperature. As the temperature increases, the adsorption efficiency of the adsorbent material decreases for all concentrations of uranium. This indicates that lower temperatures are more efficient in removing uranium from the aqueous solution.

Overall, these two graphs demonstrate the importance of controlling the temperature during the uranium removal process, as the adsorption capacity and efficiency of the developed adsorbent material are influenced by the temperature. SD of the overall adsorption measurements is ±3%.

When comparing the effectiveness of various adsorbent materials at different temperatures, it was observed that at 25 °C, the adsorbent material had a consistent effectiveness of around 90%, but its effectiveness began to drop when the uranium concentration reached 100 ppm. However, at higher temperatures (35 °C, 45 °C, and 55 °C), the effectiveness of the adsorbent material decreased as the temperature rose for uranium concentrations in the range of 40–60 ppm. Interestingly, the effectiveness improved for a uranium concentration range of 80–100 ppm. Overall, these findings suggest that the effectiveness of the adsorbent material is influenced by both temperature and uranium concentration. At 25 °C, the adsorbent material is reliable in removing uranium from a solution, while at higher temperatures, the effectiveness is more dependent on the uranium concentration.

#### 2.2.3. Effect of pH on Adsorption at Two U Concentrations (50 and 100 ppm) at Room Temperature

The effect of pH on adsorption was investigated at two uranium concentrations (50 and 100 ppm) at room temperature. The results, presented in Table 3A, indicate that at a concentration of 100 ppm, the adsorbent material had the highest effectiveness at a pH level of 6. Similarly, the experiment was repeated at a uranium concentration of 50 ppm, and the results were presented in Table 3B. It was observed that at lower uranium concentrations, the adsorbent effectiveness improved, and the highest effectiveness was observed at a pH level of 6, outperforming other pH levels. These findings suggest that pH plays a crucial role in the adsorption of uranium, and the optimal pH level for maximum adsorption varies depending on the uranium concentration. Therefore, it is important to consider both the pH level and uranium concentration when designing adsorption systems for U removal.

The Appendix A document includes two figures related to the adsorption of uranium as a function of pH level at two concentrations. Appendix A shows the equilibrium adsorption, while Appendix A presents the adsorption effectiveness %. Both figures reveal a clear trend where the effectiveness of the adsorbent material increases as the pH level increases, reaching its peak at a pH of 6. As the pH level continues to increase beyond 6, the effectiveness of the adsorbent decreases. Therefore, it can be concluded that the optimal pH level for maximum adsorption effectiveness is 6. To achieve the optimum adsorption capacity, further experiments were conducted to investigate adsorption kinetics and isotherms at the optimal pH level. These pH-dependent behaviors are consistent with previous reports in the literature on the adsorption of U(VI) on functionalized mesoporous silica materials [25,27].

#### 2.2.4. Effect of Material Weight at Room Temperature for 60 min

The weight of the adsorbent material is a critical factor in determining its adsorption effectiveness once it reaches saturation. To investigate this, several experiments were conducted on the available samples at their most effective pH, measuring the potential for U adsorption at two concentrations (50 and 100 ppm) with an adsorption duration of 60 min. For the flasks containing a U concentration of 100 ppm, as listed in Table 4A, it is clear that as the weight of the adsorbent material increased, the effectiveness of the adsorption also increased. At a weight of 50 mg, the effectiveness reached 94.57%. Notably, at a weight of 25 mg, the adsorbent material exhibited a peak improvement in its effectiveness of 89.3%, outperforming other weight values. Similarly, for flasks containing a U concentration of 50 ppm, as shown in Table 4B, it is evident that increasing the weight of the adsorbent material resulted in higher adsorption effectiveness. At a weight of 50 mg, the effectiveness reached 98.74%. However, at a weight of 25 mg, the adsorbent material showed a peak improvement in its effectiveness of 95.16%, which then dropped to 75.12% when the weight was increased.

#### 2.2.5. Effect of Mixture Solution at Room Temperature for 60 min

In this experiment, the effectiveness of the adsorbent material in targeting U compared to other heavy metals in a mixture solution was investigated at room temperature for 60 min. Firstly, a flask containing a mixture of (Cr, Ni, Cu, Zn, Cd, Pb, U) was prepared, with each element mixed with 50 mL distilled water to reach a concentration of 100 ppm. A total of 50 mg of the adsorbent material was added to the flask and placed in a water bath shaker, and after 60 min, the mixture was measured using ICP-MS. The results, as shown in Table 5A, indicate that the adsorbent material was most effective in targeting U, with an effectiveness of 58.27%. In the second experiment, a flask containing a mixture of (Cr, Ni, Cu, Zn, Cd, U) was prepared, with Pb removed from the previous mixture. Again, each element was mixed with 50 mL distilled water to reach a concentration of 100 ppm. A total of 50 mg of the adsorbent material was added to the flask and placed in a water bath shaker, and after 60 min, the mixture was measured using ICP-MS. The results, as shown in Table 5B, indicate that the adsorbent material was even more effective in targeting U in this mixture, with an effectiveness of 97.59% compared to other heavy metals.

### 2.3. Mesoporous Silica Nanoparticles—Phosphate (Diphosphate)-Modified MSNs (Phosphate (Diphosphate)-MSNs)—Novel Material (B)

The effectiveness of the adsorbent material was measured under several experimental conditions.

#### 2.3.1. Effect of Time on U Adsorption at Room Temperature

The first experiment measured the adsorption performance of the developed material over time (5, 10, 15, 30, 60, 120, 180, and 240 min) using the same procedure as the first developed adsorbent material. The measurements were taken using ICP-MS. Appendix A shows that the effectiveness of the adsorbent material increased with time. The highest effectiveness was observed at 120 min, indicating that this time is optimal for the developed adsorbent’s best performance.

#### 2.3.2. Effect of Temperature on U Adsorption

The same procedure as the first developed adsorbent material was used in this experiment. Results of flasks with different U concentrations at 25 °C for 60 min showed that as U concentrations increased, the adsorbent’s effectiveness decreased, as measured by ICP-MS. At 25 °C, Appendix A shows that the adsorbent effectiveness was 73.81% at 10 ppm of U, but decreased to 31.31% at 100 ppm of U, indicating that the adsorbent material is more effective at lower U concentrations.

At 35 °C, Appendix A shows that the adsorbent effectiveness was 49.59% at 10 ppm of U and 74.35% at 100 ppm of U. The average adsorbent effectiveness was higher compared to the 25 °C experiment, but dropped at higher U concentrations, suggesting that the adsorbent material is more effective at lower U concentrations. At 45 °C, Appendix A shows that the adsorbent effectiveness was 87.53% at 10 ppm of U but dropped to 36.95% at 100 ppm of U. Compared to the 35 °C experiment, the adsorbent effectiveness slightly improved. At 55 °C, (Appendix A) shows that the adsorbent effectiveness was 86.70% at 10 ppm of U, but dropped to 46.97% at 100 ppm of U. The results were similar to those of the 45 °C experiment. Appendix A shows the influence of temperature on equilibrium adsorption qt (mg/g) as a function of U concentration. Appendix A shows the influence of temperature on adsorption effectiveness % as a function of U concentration. The figure clearly shows that as the U concentration increases, the effectiveness of the adsorbent material decreases until the concentration reaches 40–60 ppm, and the effectiveness of the adsorbent material improves as the concentration increases. However, at the 55 °C experiment, the effectiveness decreased at concentration 80–100 ppm due to experimental error.

#### 2.3.3. Effect of pH on Adsorption in Two Concentrations of U (50, 100 ppm) at Room Temperature

The effect of pH on the adsorption reaction of the second developed adsorbent material was investigated at two concentrations of U (50 and 100 ppm) at room temperature using the same procedure as the first material. The measurements were taken using ICP-MS. The results in Appendix A show that the highest adsorbent effectiveness at 100 ppm is achieved at a pH level of 6. The experiment was repeated at 50 ppm, as shown in Appendix A, which reveals that the highest effectiveness of the adsorbent material is achieved at a pH level of 4 compared to other pH levels. Appendix A illustrates the equilibrium adsorption as a function of pH level at two U concentrations, while Appendix A depicts the adsorption effectiveness % as a function of pH level at two U concentrations. The optimum pH level for a U concentration of 100 ppm is 6, whereas for a U concentration of 50 ppm, it is 4. The changes in the results compared to the first material may be due to a faulty input to the ICP-MS.

#### 2.3.4. Effect of Material Weight at Room Temperature for 60 min

The U adsorption potential of the available samples at their most effective pH was measured at two concentrations of U (50 and 100 ppm) for 60 min using ICP-MS. The same procedure as the first material was used to investigate the effect of material weight at room temperature for 60 min. The results in Appendix A show that as the weight of the adsorbent material increased for flasks containing a U concentration of 100 ppm, the effectiveness of the adsorption increased too. Almost all of the weight results showed similar adsorbent material effectiveness, with the highest being at 30 mg, which reached 38.4%. The results in (Appendix A) revealed a similar trend for flasks containing a U concentration of 50 ppm. As the weight of the adsorbent material increased, the effectiveness of the adsorption increased too, with almost all weight results showing similar adsorbent material effectiveness. The highest effectiveness was at 30 mg, reaching 38.74%.

#### 2.3.5. Effect of Mixture Solution at Room Temperature for 60 min

In this experiment, the effectiveness of the adsorbent material in targeting U compared to other heavy metals was measured using a mixture solution at room temperature for 60 min. Firstly, a flask was prepared with a mixture of Cr, Ni, Cu, Zn, Cd, Pb, and U, each mixed with 50 mL distilled water to reach 100 ppm concentration. Then, 50 mg of the adsorbent material was added to the flask and placed in a water bath shaker. After 60 min, the effectiveness of the adsorbent material on the mixture was measured by ICP-MS. The results showed that the adsorbent material mostly targeted U compared to the other available heavy metals, reaching an effectiveness of 40% as shown in Appendix A. Secondly, a flask was prepared with a mixture of Cr, Ni, Cu, Zn, Cd, and U, where Pb was removed from the first mixture. Each element was mixed with 50 mL distilled water to reach 100 ppm concentration, and 50 mg of the adsorbent material was added to the flask and placed in a water bath shaker for 60 min. The effectiveness of the adsorbent material in targeting U among other available heavy metals was measured by ICP-MS, and the results showed that the adsorbent material specifically targeted U, reaching an effectiveness of 37%, as shown in Appendix A.

## 3. Methodology

### 3.1. The Materials

The materials used were similar to those published by Abdullah M. Alswieleh et al. in 2021, with minor differences [28]. Deionized water was used, which was acquired using an Elga Pure Nanopore 18.2 MΩ. The following and all other chemicals were purchased from Sigma-Aldrich (St. Louis, MI, USA): 3-Aminopropyltriethoxysilane (APTES, >98%), N-Cetrimonium Bromide (CTAB, 98%), Tetraethyl orthosilicate (TEOS, 98%), (3-Glycidyloxypropyl)trimethoxysilane (GPTMS) (98%), O-Phosphorylethanolamine (98%), Hexane (HPLC Grade), Methanol (ultrapure, HPLC Grade, 99.8%), Ethanol (99.8%, HPLC grade), Toluene (ACS Reagent Grade), and Ammonium hydroxide (28 wt. %). The dimethyl sulfoxide (DMSO) and hydrochloric acid (HCl) were acquired from Fisher Scientific: Lab Equipment and Lab Supplies. The doubly charged uranium (U^++)^ 900 ppm solution used in this study was obtained using Uranyl nitrate hexahydrate (Aldrich, USA). The elements (Cr, Ni, Cu, Zn, Cd, Pb) used in the study were acquired from Merck, Darmstadt, Germany. Chromium(III) nitrate nonahydrate; nickel(II) nitrate hexahydrate; copper(II) nitrate tri-hydrate; zinc nitrate hexahydrate; cadmium nitrate tetrahydrate; and Lead(II) nitrate were used to obtain (Cr), (Ni), (Cu), (Zn), (Cd), and (Pb), respectively. All chemicals were used as received.

### 3.2. The Synthesis

Samples containing UO^++^ with different concentrations were prepared using three 500 mL containers. The UO^++^ was extracted from a 900-ppm solution using proportional sets and the equation C1V1 = C2V2 to calculate equivalent amounts, strengths, and substitutes. The first container contained 10 ppm, which was equivalent to 5.55 mL of the 900 ppm UO^++^ solution; the second container contained 50 ppm, which was equivalent to 27.77 mL of the 900 ppm UO^++^ solution; and the third container contained 100 ppm, which was equivalent to 55.55 mL of the 900 ppm UO^++^ solution. All the containers were then mixed with distilled water to a total volume of 500 mL.

#### 3.2.1. Synthesis of Mesoporous Silica Nanoparticles

The synthesis of mesoporous silica nanoparticles was carried out in a similar manner as described by Abdullah M. Alswieleh et al. in 2019. One gram of CTAB was dissolved in 160 mL of deionized water under mixing. Then, 7 mL of concentrated ammonia water (28%) was added to create a clear solution. A mixture of 20 mL n-hexane and 5 mL TEOS was slowly added to the solution within 30 min under continuous mixing. As the reaction continued at 35 °C, a consistent creamy colloidal solution was slowly formed under non-stop mixing at around 200 rpm. After mixing for half a day, the product was collected by centrifugation, washed with deionized water and ethanol, and then dried in an oven at 100 °C for 2 h [27,29]. The sample was then subjected to solvent removal treatment to remove the CTAB templates. For this, 1.5 g of the sample was re-dissolved in 160 mL methanol and a strong aqueous solution of HCl (12 M, 9 mL) was added to the mixture, which was then heated under reflux for one day. The solvent extraction was repeated twice, and the sample was collected by centrifugation, washed with ethanol 7 times, and finally dried for 12 h.

#### 3.2.2. Synthesis of 3-Glycidyloxypropyl-functionalized MSNs (Epo-MSNs)

To synthesize 3-glycidyloxypropyl-functionalized mesoporous silica nanoparticles (Epo-MSNs), we first achieved a 3-glycidyloxypropyl-coated mesoporous silica surface. To do this, we suspended 1.5 g of nanoparticles in a mixture of 2.5 mmol of (3-glycidyloxypropyl)trimethoxysilane in 50 mL of dry toluene. We then heated the resulting mixture under reflux for one day. Afterward, we collected the coated nanoparticles through centrifugation, washed them twice with toluene, five times with ethanol, and then dried them under vacuum. 

#### 3.2.3. Preparation of Phosphate Modified MSNs (Phosphate-MSNs)

To prepare phosphate-modified mesoporous silica nanoparticles (phosphate-MSNs), we first added 1 g of 3-glycidyloxypropyl-coated MSNs to a round bottom flask containing 20 mL of DMSO. We then heated the reaction mixture to 60 °C for 1 h. Next, we added a solution of O-phosphorylethanolamine (3 g) and water (10 mL) to the reaction mixture and stirred it for 48 h at 85 °C. After the reaction was complete, we collected the phosphate-MSNs using centrifugation. The solid material was then washed with deionized water and methanol, and finally dried in an oven overnight. Phosphate-MSNs have many potential applications, particularly in biomedicine. The phosphate group on the MSNs surface can bind to uranium ions, which could facilitate bone tissue regeneration. Overall, this method provides a straightforward way to prepare phosphate-modified MSNs with desirable surface properties for biomedical applications. By adjusting the reaction conditions, such as the reaction temperature and time, we can optimize the properties of the resulting phosphate-MSNs for specific uses.

#### 3.2.4. Preparation of 3-Aminopropyl-functionalized MSNs (AP-MSNs)

To prepare 3-aminopropyl-functionalized mesoporous silica nanoparticles (AP-MSNs), we first suspended 1.5 g of the previously prepared nanoparticles in a solution of 2.5 mmol of aminopropyltriethoxysilane (APTES) in 50.0 mL of dry toluene. We then heated the reaction mixture to reflux for 24 h. After the reaction was complete, we collected the AP-MSNs using centrifugation. We then washed the particles five times with water and five times with methanol to remove any unreacted APTES or other impurities. Finally, we dried the AP-MSNs in an oven. AP-MSNs have many potential applications, particularly in biomedicine. The amine groups on the MSNs surface can be used to covalently attach a variety of biomolecules, such as proteins or DNA, allowing for targeted drug delivery or sensing applications. Additionally, the nanoparticles could be used for imaging or as supports for catalysts. Overall, this method provides a simple and effective way to functionalize MSNs with amino groups, which can be further modified for a variety of applications. By adjusting the reaction conditions, such as the APTES concentration or reaction time, we can tailor the properties of the resulting AP-MSNs for specific uses.

#### 3.2.5. Preparation of ATRP Initiator Immobilization, BiBB-MSN

The ATRP initiator-functionalized MSN was prepared as follows: 1 g of AP-MSNs was mixed with 30 mL of dichloromethane and 1.5 mL of triethylamine; then, 1.2 mL of 2-bromo-2-methylpropionyl bromide (BiBB) was added to 5 mL of DCM and slowly introduced dropwise into the reaction mixture at room temperature, which was stirred for 24 h; the resulting product was mixed with 25 mL of DCM using sonication, filtered, and washed with dichloromethane and ethanol; finally, the product was dried in an oven.

#### 3.2.6. Preparation of Glycidyl Methacrylate (GMA) Brushes Grafted GMA-MSNs

A total of 200 mg of BiBB-MSN (from the previous step) was added to a mixture of isopropanol (4.0 mL) and water (1.0 mL), and the reaction mixture was sonicated for 30 min. Next, 2.7 g of glycidyl methacrylate (GMA) was added to the mixture. The reaction mixture was then left at 40 °C under a nitrogen atmosphere for 30 min. Afterwards, a mixture of 30 mg of 2,2′-bipyridine and 2.2 mg of Cu(II)Br2 was added to the reaction mixture under a nitrogen atmosphere for 15 min. Then, 2 mg of Cu(I)Cl was added to the reaction mixture. Finally, the polymerization of glycidyl methacrylate on the MSNs surface was left for 2 h at 40 °C under a nitrogen atmosphere. The resulting product was collected by centrifugation, washed with water and methanol five times, and dried in an oven.

#### 3.2.7. Preparation of Polyphosphate-MSNs

GMA-MSNs (1 g) were added to a round bottom flask containing 20 mL of DMSO and heated to 60 °C for 1 h. O-Phosphorylethanolamine (3 g) in 10 mL of water was then added to the reaction mixture, which was stirred for 48 h at 85 °C. Polyphosphate-MSNs were separated by centrifugation, washed five times with deionized water and methanol, and dried in an oven at 50 °C overnight.

It is indicated in Figure 7 that the synthesis of MSNs involved several steps. The first step involved removing the organic surfactant from the mesoporous-nanoparticle, which was followed by the attachment of (3-glycidyloxypropyl)trimethoxysilane onto the inner and outer surfaces of the mesoporous-nanoparticles. The epoxy groups on the nanoparticles were then reacted with O-Phosphorylethanolamine, leading to the grafting of phosphate functional groups onto the inner and outer surfaces of the mesoporous silica nanoparticles using surface-initiated atom transfer radical polymerization (SI-ATRP). The morphology of both phosphate-MSNs and poly-phosphate-MSNs was analyzed using scanning electron microscopy (SEM).

### 3.3. Measurement and Characterization

The successful preparation and modification of the mesoporous silica nanoparticles were confirmed using several techniques. The surface area of the nanoparticles was analyzed by nitrogen physisorption isotherms using a Gemini 2375 Surface Area BET Analyzer. Prior to analysis, the samples were heated under vacuum for 150 min at 150 °C. The Brunauer–Emmett–Teller (BET) method was used to analyze the surface area using experimental points at a pressure (P/Po) of 0.05–0.25. The total pore volume (Wo) was measured from the N_2_ amount adsorbed at the P/Po of 0.99 for every sample. Infrared spectroscopy of all samples was conducted using a Nicolet™ iS™ 10 Fourier transform infrared spectroscopy (FTIR) spectrometer in KBr pellets in the 4000 to 400 per cm (cm^−1^) region. Thermogravimetric analysis was conducted by SII TGA 6300 instrument under a nitrogen atmosphere with a heating rate of 10 °C/min.

The images of the mesoporous silica nanoparticles were obtained using a JSM-6380 (JEOL) SEM. The samples were directly analyzed without any preparation or modification, with an accelerating voltage of 5 kV. The performance of the synthesized materials for uranium removal was evaluated using inductively coupled plasma mass spectrometry (ICP-MS). To prepare the supernatant samples, 0.4 mL of concentrated nitric acid (HNO₃) was added, and the total volume was adjusted to 10 mL using deionized water before conducting the ICP-MS analysis. 

The experiments were carried out under various conditions, which are set out below.

#### 3.3.1. Effect of Time on U Adsorption at Room Temperature

A total of 20 mg of the adsorbent material was mixed with 50 mL of an initial U concentration of 100 ppm at pH 6 at room temperature (25 °C) in eight flasks. The flasks were then placed in a water bath shaker to maintain a constant temperature and to keep the contents mixed. The flasks were analyzed based on the assigned time.

#### 3.3.2. Effect of Temperature on U Adsorption

A total of 20 mg of the developed adsorbent was mixed with 50 mL of different U concentrations (10, 40, 60, 80, 100 ppm) in five flasks. The mixture was then mixed for 60 min in a water bath shaker at 25 °C, followed by repeating the steps at 35, 45, and 55 °C.

#### 3.3.3. Effect of pH on U Adsorption in Two Concentrations of U (100, 50) at Room Temperature

Eight flasks were prepared, and 20 mg of the developed adsorbent material was placed in each flask. Four of the flasks were filled with U concentration of 100 ppm at different pH levels (pH 2, 4, 6, 8) by adding chemical substances. The flasks were then placed in a water bath shaker at room temperature for 60 min.

#### 3.3.4. Effect of Material Weight at Room Temperature for 60 min

Twelve flasks were prepared to contain different U concentrations (50, 100 ppm) with varying weights of the developed adsorbent material (10, 15, 20, 25, 30, 50 mg). The flasks were placed in a water bath shaker at room temperature for 60 min.

#### 3.3.5. Effect of Mixture Solution at Room Temperature for 60 min

Two flasks were prepared to contain mixtures of (Cr, Ni, Cu, Zn, Cd, Pb, U) and (Cr, Ni, Cu, Zn, Cd, U), respectively. Each element was mixed with 50 mL of distilled water to reach a concentration of 100 ppm. Then, 50 mg of the adsorbent material was added to each flask, and the flasks were placed in a water bath shaker at room temperature for 60 min.

All measurements were conducted using ICP-MS. Each prepared flask from all experiments was transferred to a water bath shaker and then to a 50 mL bottle, which was placed in a centrifuge by Hettich ROTINA 35 (1000 revolutions per minute) for 10 min. Then, 9 mL of the sample was taken from the 50 mL bottle and mixed with 1 mL of HNO₃ (1%) before being transferred to the ICP-MS.

## 4. Conclusions

The effects of time, temperature, pH, material weight, and mixture solution on the adsorption capacity of the novel adsorbent material, phosphate (diphosphate)-modified MSNs (phosphate (diphosphate)-MSNs), in the removal of uranium (U) from aqueous solutions were investigated. The phosphate (diphosphate)-MSNs exhibited excellent performance under all testing conditions, surpassing expectations prior to its development. The highest removal of 85.35% was achieved after 120 min at room temperature, indicating the favorable effect of time on U adsorption. At a U concentration of 10 ppm and a temperature of 25 °C, the highest removal of 96.7% was obtained, highlighting the influence of temperature on adsorption efficiency. A pH of 6 and a U concentration of 50 ppm resulted in the highest removal of 91.89%, indicating the significance of pH in adsorption. Moreover, a material weight of 25 mg and a U concentration of 50 ppm led to the highest removal of 95.16%, emphasizing the role of material weight. The phosphate (diphosphate)-MSNs exhibited a high removal efficiency of 58.27% when targeting U in the mixture at room temperature for 60 min, showcasing their effectiveness in the presence of mixed solutions. These results demonstrate the exceptional efficiency of phosphate (diphosphate)-modified MSNs (phosphate (diphosphate)-MSNs) in the removal of uranium from aqueous solutions. The findings indicate the potential of this novel adsorbent material for practical applications in treating uranium-contaminated water sources. Further research is needed to optimize the synthesis process and explore its effectiveness under different environmental conditions.

## Figures and Tables

**Figure 1 molecules-28-05642-f001:**
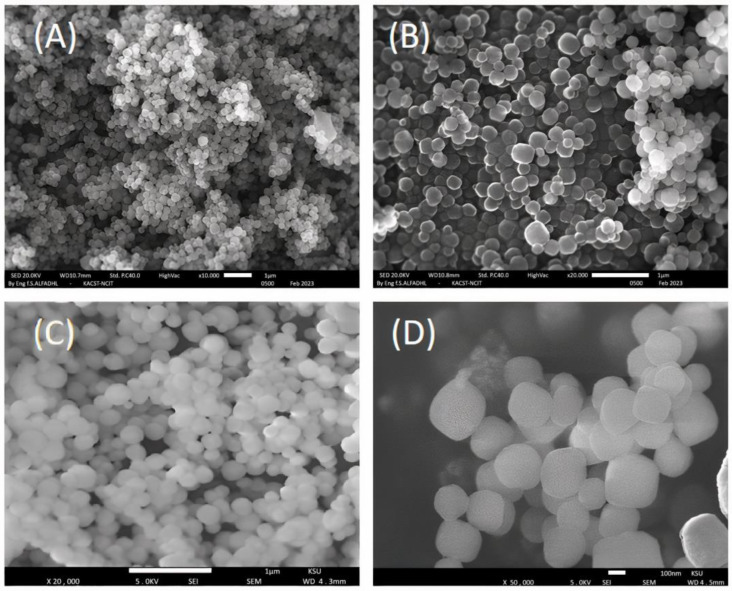
Images of prepared mesoporous silica nanoparticles using SEM technique: (**A**) unmodified MSNs (magnification 1 µm); (**B**) phosphate-modified silica nanoparticles (magnification 1 µm); (**C**) unmodified MSNs (magnification 1 µm); (**D**) polyphosphate-modified silica nanoparticles (magnification 100 nm).

**Figure 2 molecules-28-05642-f002:**
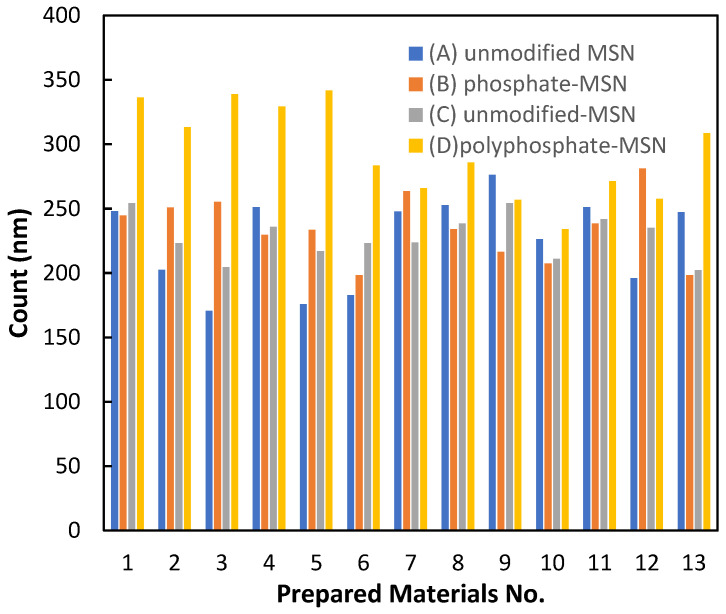
Particle size diameter distribution charts for prepared materials.

**Figure 3 molecules-28-05642-f003:**
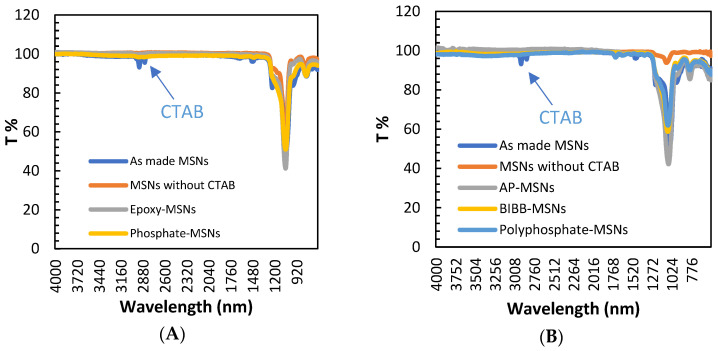
FTIR data for synthesized material: (**A**) phosphate-MSNs, (**B**) polyphosphate-MSNs.

**Figure 4 molecules-28-05642-f004:**
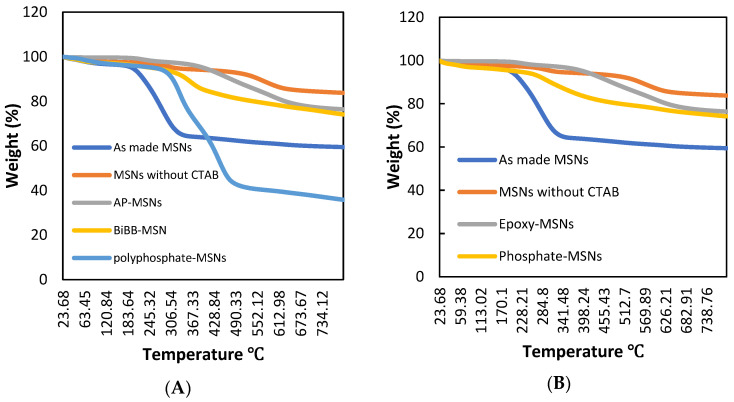
TGA data for synthesized material; (**A**) phosphate-MSNs, (**B**) polyphosphate-MSNs.

**Figure 5 molecules-28-05642-f005:**
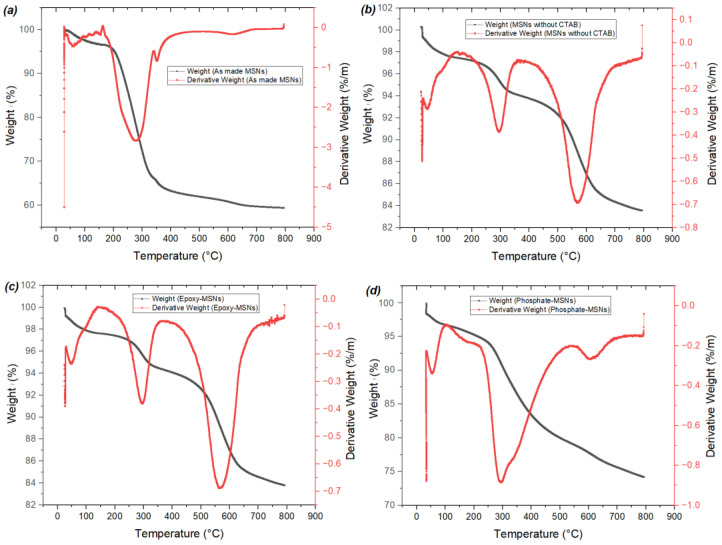
TGA and DTG thermograms illustrating the correlation between mass loss rate and temperature for material (A) phosphate-MSNs: (**a**) as-made MSNs; (**b**) MSNs without CTAB; (**c**) epoxy-MSNs; (**d**) Phosphate-MSNs.

**Figure 6 molecules-28-05642-f006:**
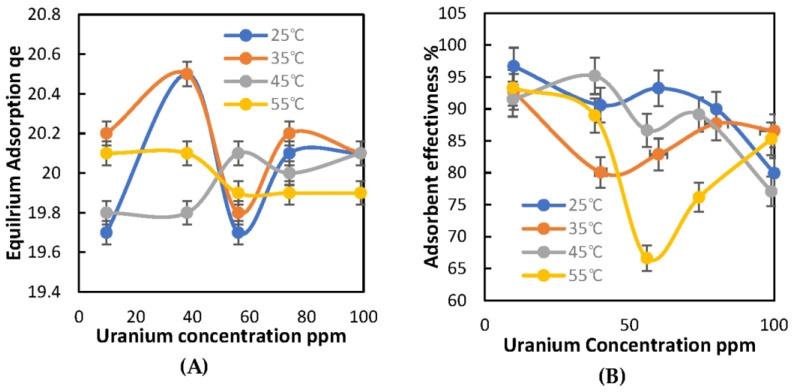
The Influence of temperature on the adsorbent: (**A**) equilibrium adsorption qe (mg/g) against U concentration; (**B**) adsorption effectiveness % as a function of U Concentration ppm.

**Figure 7 molecules-28-05642-f007:**
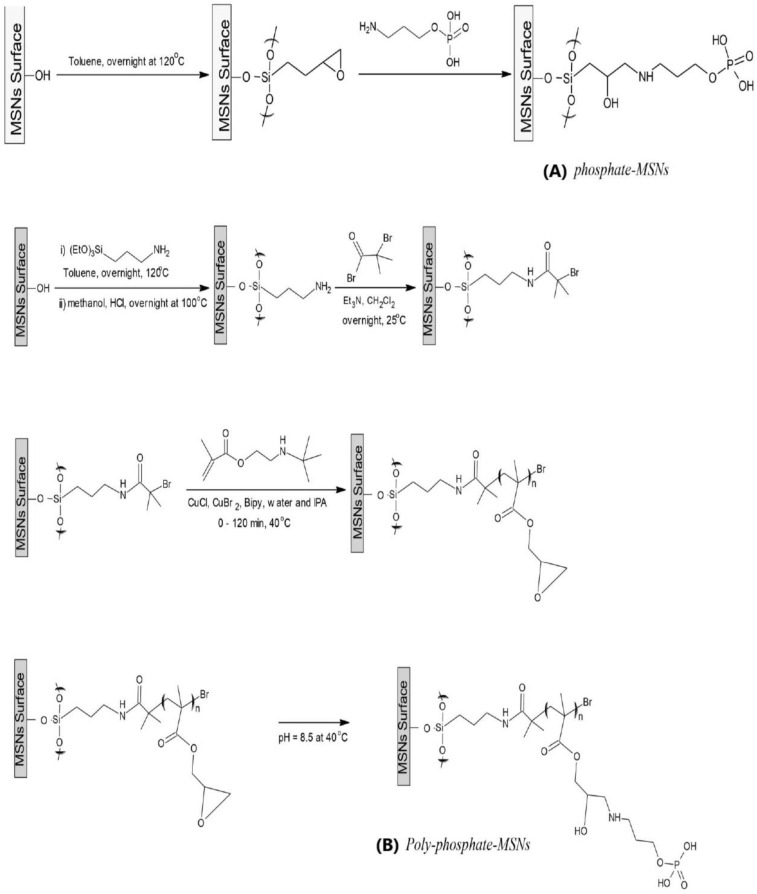
Surface adjustment of mesoporous silica nanoparticles (MSNs) with: (**A**) phosphate-MSNs, (**B**) polyphosphate-MSNs.

**Table 1 molecules-28-05642-t001:** Physiochemical data for the phosphate-MSNs and polyphosphate-MSNs samples.

Material	BET Surface-Area (m^2^.g^−1^)	Pore-Volume (cm^3^.g^−1^)
MSNs	1048	1.51
phosphate-MSNs	570	0.95
Polyphosphate-MSNs	10.02	0.0166

**Table 2 molecules-28-05642-t002:** Effect of time and temperature on U adsorption (material (A)).

**A. Effect of Time on U Adsorption at Room Temperature**				
**U^++^ Concentration (ppm)**	**t (Minute)**	**Ce (mg/L)**	**%**	**Qt (mg/g)**
100	5	29.78	70.22	20.1
	10	23.17	76.83	20.1
	15	28.79	71.21	20.2
	30	22.04	77.74	19.9
	60	22.61	77.16	20.1
	120	14.5	85.35	20.2
	180	16.53	83.3	20
	240	14.27	85.59	20.1
**B. Effect of temperature on U adsorption (COF) at 25 °C.**				
**U^++^ concentration (ppm)**	**Ce (mg/L)**	**%**	**qe (mg/g)**	**Ce/qe**
10	0.32	96.7	19.7	0.02
40	3.58	90.58	20.5	0.17
60	3.77	93.27	19.7	0.19
80	7.43	89.96	20.1	0.37
100	19.87	79.93	20.1	0.99
**C. Effect of temperature on U adsorption (COF) at 35** **°C.**				
**U^++^ concentration (ppm)**	**Ce (mg/L)**	**%**	**qe (mg/g)**	**Ce/qe**
10	0.71	92.68	20.2	0.04
40	7.57	80.08	20.5	0.37
60	9.58	82.89	19.8	0.48
80	9.09	87.72	20.2	0.45
100	13.3	86.57	20.1	0.66
**D. Effect of temperature on U adsorption (COF) at 45** **°C.**				
**U^++^ concentration (ppm)**	**Ce (mg/L)**	**%**	**qe (mg/g)**	**Ce/qe**
10	0.82	91.55	19.8	20
40	1.83	95.18	19.8	20
60	7.47	86.66	20.1	20
80	8.05	89.12	20	20
100	22.69	77.08	20.1	20
**E. Effect of temperature on U adsorption (COF) at 55** **°C.**				
**U^++^ concentration (ppm)**	**Ce (mg/L)**	**%**	**qe (mg/g)**	**Ce/qe**
10	0.65	93.3	20.1	0.03
40	4.19	88.97	20.1	0.21
60	18.69	66.63	19.9	0.94
80	17.64	76.16	19.9	0.89
100	14.53	85.32	19.9	0.73

Notes: Ci (mg/L) = initial concentration; qe (mg/g) = equilibrium adsorption. SD of the overall adsorption measurements is ±3%.

**Table 3 molecules-28-05642-t003:** Effect of pH on adsorption at two U concentrations (50 and 100 ppm) at room temperature.

**A. The Effect of pH at Concentration of U (100 ppm) at Room Temperature.**				
pH	Ci (mg/L)	Ce (mg/L)	%	qe (mg/g)
2	100	87.3	12.7	19.9
4		56.11	43.89	20
6		27.15	72.85	20.1
8		44.06	55.94	19.9
**B. The effect of pH at concentration of U (50 ppm) and room temperature.**				
pH	Ci (mg/L)	Ce (mg/L)	%	qe (mg/g)
2	50	44.37	11.26	20.1
4		24.38	51.24	20.1
6		4.057	91.89	19.9
8		11.95	76.1	20.1

**Table 4 molecules-28-05642-t004:** Effect of material weight at room temperature for 60 min (U concentration 50, 100 ppm) (Material A).

**A. 60 min at U Concentration 100 ppm.**			
**Weight (mg)**	**Ce (mg/L)**	**%**	**qe (mg/g)**
10	46.37	53.63	10.1
15	36.7	63.3	15
20	34.94	65.06	20.1
25	10.7	89.3	25
30	16.42	83.58	29.9
50	5.43	94.57	49.9
**B. 60 min at U concentration 50 ppm.**			
**Weight (mg)**	**Ce (mg/L)**	**%**	**qe (mg/g)**
10	27	46	10
15	17.63	64.74	15
20	18.86	62.28	19.9
25	2.42	95.16	25.1
30	12.44	96.12	29.9
50	0.63	98.74	49.9

**Table 5 molecules-28-05642-t005:** Effect of mixture solution at room temperature for 60 min (Material A).

**A. The Effect of Adsorption with Pb.**				
**Element**	**Ci (mg/L)**	**Ce (mg/L)**	**%**	**qe (mg/g)**
Cr	100	77	23	49.9
Ni	100	60.19	39.81	
Cu	100	88.69	11.31	
Zn	100	93	7	
Cd	100	99	1	
Pb	100	76	24	
U	100	41.73	58.27	
**B. The effect of adsorption in the absence of Pb.**				
**Element**	**Ci (mg/L)**	**Ce (mg/L)**	**%**	**qe (mg/g)**
Cr	100	41	59	50.1
Ni	100	30.18	69.82	
Cu	100	44.52	55.48	
Zn	100	46.53	53.47	
Cd	100	50	50	
U	100	2.41	97.59	

## Data Availability

Data and samples of the material are available from the authors.

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
