# Peer review of "Treatment of Uranium-Contaminated Ground Water Using Adsorption Technology via Novel Mesoporous Silica Nanoparticles"

_molecules, 2023, doi:10.3390/molecules28155642_

Round 1

Reviewer 1 Report

Manuscript ID: molecules-2337688

Abdulmalik S. Alshammari and co-authors reported "Treatment of Contaminated Underground Water from Uranium Using Adsorption Technology by Novel Mesoporous-silica Nanoparticles" Although the topic is interesting, but some important aspects were not performed. Following comments should be addressed before possible consideration for publication in worthy Journal of Molecules.

1.      In abstract various abbreviations were used like SI-ATRP. Use full form when write 1st time and then abbreviation be used throughout the manuscript.

2.      In introduction more literature should be reviewed and some latest adsorbents should be discussed here to enhance the novelty of work like, https://doi.org/10.1080/03067319.2021.1998471. Surfaces and Interfaces 34 (2022) 102324, Applied Clay Science 190 (2020) 105564.

3.      Headings should not include in Introduction like Overview and Background & Problem Statement

4.      The Materials should not be in bullet. Write them in paragraph

5.      Graphical representation of synthesis scheme should be added

6.      Synthesis method should be divided into sub headings

7.      Fig.1 is not in proper way, some portion is missing, provide its best.

8.      Fig 4 &5 temperature should be mentioned in ℃.

9.      Format of paper is not good, a lot of changings are required to set graphs as well as data & tables

10.  Figures of BET analysis should be procvided

11.  Merge table 13 to 17 into one table

12.  Merge table 2 to 6 into one table

13.  Merge table 7 to 8 into one table

14.  Merge table 9 to 10 into one table

15.  Merge table 11 to 12 into one table

16.  Merge table 20 to 21 into one table

17.  Merge table 22 to 23 into one table. Also shift less important tables anf figures in supporting information file

18.  No of figures is much high. Figures should not be more than 6-7. Move less important in supporting information file. Also combine figures (4 into one) just like Fig.2

19.  XRD analysis should be done to show synthesized material structure and morphology.

20.  Captions of tables and figs. are repeated like above & below of figs, it should be avoided.

21.  Reusability and stability should be checked

22.  Proposed mechanism of adsorption should be discussed in details

23.  There are so many typo grammatical errors in whole manuscript, should be revised by some native speaker and formatting should be checked.

Reviewer 2 Report

Dear Authors,

The study entitled "Treatment of Contaminated Underground Water from Radioactive Material Using Adsorption Technology by Novel Mesoporous-silica Nanoparticles" aims to demonstrate the modern nanoparticles which could be useful for the absorption of the radioactive materials from the underground water.

Fabrication of such materials is very important nowadays, and could be useful not only for usual water treatment (water conditioning), but also in case of technogenic accidents.

In spite of the seeming completeness, this study must be improved and corrected. Key points are listed below.

1. p. 2, Introduction

The Authors write in detail previous investigations related to the underground water analysis. Thus, the study of "Shabana & Kinsara et al." takes up much space in this manuscript. In my opinion this information should be presented in narrative format. Moreover, the information could be demonstrated in graphical or tabular form, which helps to improve the readability and visual attractivity of the paper.

2. Please, use the same units throughout the text, e.g.. Bq/L (p. 2, line 53), 5.55 mL (p. 4, line 176), 27.77 mL (p. 4, line 177), 30 mL (p. 5, line 215) etc. I kindly recommend the Authors to carefully check the text before submission.

3. The same note is related to subscript and superscript symbols and digits. Please, check each chemical formula (e.g. p. 3, line 126), degree symbols (p. 4, line 189, p. 5, line 224, p. 7, line 272, p. 7m line 302, etc.).

4. p. 3, line 131

Please, interpret the "DEHCEBP".

5. p. 4, line 172

I kindly ask the Authors to clarify the writing "UO++" using superscript or/and subscript symbols. Also the chemicals used for the solutions should be provided at the Materials section. Apart from the U, I recommend the Authors to include also the detailed information of Cr, Ni, Cu, Zn, Cd, Pb chemicals which were used in the study.

6. p. 4, line 197

Please, correct misprint "misxtue". It should be written as "mixture".

I kindly recommend the Authors to check in detail the text for the purpose of avoiding mistakes.

7. p. 5, line 234

The temperature and duration of drying should be provided.

8. Figure 1, p. 6

The figure is cut. Moreover, the "A" and "B" are not included in the figure, while the figure caption contains these designations.

9. p. 6, subsection 2.3

The measurement methods should be divided into individual subsections.

At the same time, I kindly recommend to include additional information into this subsection, e.g. applied voltage, working distance, and detector type in SEM analysis, etc. 

10. p. 8, line 307

Please, delete "Authors"

11. p. 8, Figure 2

Please, provide the magnification of the SEM microphotographs.

Moreover, the data related to the size distribution should be demonstrated: histograms, tables with the means, SD, max, and min sizes of the nanoparticles. The quantity of the particles measured also should be provided.

12. P. 8, lines 325, 329, p. 9, lines 332-335

Superscript symbols must be used.

13. P. 9, Figures 3 and 4

The figures must be corrected. Firstly, the IR-spectra is usually plotted in reverse order (from 4000 to 500). Secondly, the normalization of the curves should be done. Thirdly, the key peaks should be demonstrated directly on the graph.

14. p. 10, Figures 5 and 6

Derivative thermogravimetric (DTG) curves should be provided in addition to the TGA data.

15. DSC analysis is indly recommended complementary to the TGA analysis.

16. From p. 11

The Authors provided too many same-type tables with data.

Firstly, there is no SD data provided. How many times the Authors measured the concentration and absorption?

Secondly, there are a lot of tables that lead to the overloading of the text, and could confuse the readers. The Authors should make a full revision of this subsection. As a possible alternative, the tables could be transferred into the Supplement document, and the plots could be retained. At the same time the detailed descriptions must follow each figure. In current form the descriptions seem to be as "statement of fact" without necessary analysis and summary.

Turning to the tables, please, carefully check the table headers.

17. Figures 7, 8, 9, 10, 11, 12, 13, 14

Please, add the SD into each curve. In current form the data carry little conviction.

Is it possible to include additional points on the plots, e.g. uranium concentrations, pH levels? Current figures have the lack of points and the plotted functional connections should be checked and corrected.

18. It would be preferable if the Authors compare the results obtained with the other absorbents or nanoparticle systems. As a possible variant I recommend to compare with the chitosan, which is known as a very effective biosorbent of high-density metals.

19. I kindly recommend the Authors to use MDPI (ACS Style) for the references as it highlighted in the MDPI template.

Taking into account above mentioned comments, I recommend to reconsider this manuscript after major revision.

Reviewer 3 Report

The manuscript by Alshammari et al. attempts to synthesize a novel MSN using surface-initiated atom transfer radical polymerization (SI-ATRP) and test it for uranium removal in various conditions. The MSN adsorbent exhibited high removal rates, such as 85.35% at 120 minutes, with optimal results at 25°C, pH 6, and a uranium concentration of 50ppm. The authors suggest that these findings demonstrate the effectiveness of MSNs for uranium removal under different conditions. The manuscript requires several revisions/ modifications.  

Major Comments:

-          Line 57-133: In the introduction, the detail showcasing the contributions of other researchers is superfluous. It adds nothing to the article and should be deleted or moved to the discussion. Lines: 175-182: UO++ is not defined previously

-          Line 207: The authors suggest Epo-MSN can be used for targeted drug delivery. The authors fail to mention what these particles are targeting. With the current method of synthesis and purpose of synthesis, this cannot be used in vivo. Furthermore, the existing chemical makeup of these particles and extraction procedures would not make them biocompatible.

-          The authors should specify the length of the functionalized particles and the number of functionalize molecules per nanoparticle (the targeting moiety).

-          Line 307-329: The authors test the particles for effects of pH, material weight, mixture etc using 100ppm of U concn. Why was this particle concentration selected?

-          Line 336: What the is the value of polydispersity? The authors only mention the range.

-          Line 355: The authors fail to provide an explanation for the decrease in surface area and pore volume AFTER surface modification? 

-          Figure 5-6: Better quality images would be required before publication

-          Line 418: For water purification, 120 minutes would be long time. How do the authors suggest these particles be used for mass purification? Discuss.

-          Line 439-455: SD is mentioned in range instead of ± absolute value

-          Figure 7: No error bars

Overall, the article would benefit from extensive formatting and grammatical reformation.  To further clarify the results and how these particles might be physically employed for water filtration, the conclusion and discussion should be separated.

Round 2

Reviewer 1 Report

Accept

Reviewer 2 Report

Dear Authors,

Thank you very much for the submission of the revised manuscript.

In spite of the corrections made, I detected several issues, which must be takeт into consideration.

1. Line 65

Please, correct the reference [Error! Reference source not found.]

2. Line 68 and line 119

Please, be careful with the superscripts, e.g. 33 km2, and subscripts, e.g. BaSO4 or MnO2. This recommendation was included in my previous report and I kindly ask the Authors to be more attentive.

3. Line 335

The Authors noted that "the MSNs had a nano spheres shape with a polydispersity of the particle size range of 150 to 240 nm". I kindly recommend adding the diameter distribution diagrams. The diameters could be measured using SEM images and any software (e.g. ImageJ).

4. Line 363, Figure 3

It is very strange that "DTG curves are not available to authors". The instrument used (SII TGA 6300) allows to give TGA data and DTG data at the same time. Please, check it.

5. Figure 5 (Manuscript), Figures 1-6 (supplementary)

Additional (intermediate) points must be included. Moreover, SD should be demonstrated.

Unfortunately, I can not recommend the manuscript for publication in current form.

Reviewer 3 Report

Adequate corrections were made